# Tuning HAuCl_4_/Sodium Citrate Stoichiometry to Fabricate Chitosan-Au Nanocomposites

**DOI:** 10.3390/polym14040788

**Published:** 2022-02-17

**Authors:** Luis R. Torres-Ferrer, José M. López-Romero, Juan Mendez-Nonell, Maria J. Rivas-Arreola, Marisa Moreno-Ríos, Erika O. Ávila-Dávila, Evgeny Prokhorov, Yuriy Kovalenko, Diana G. Zárate-Triviño, Javier R. Revilla-Vazquez, Marco A. Meraz-Rios, Gabriel Luna-Barcenas

**Affiliations:** 1Nanosciences & Nanotechnology Program, Cinvestav Zacatenco, Ciudad de Mexico 07360, Mexico; luisruben.torres@cinvestav.mx; 2Cinvestav Querétaro, Querétaro 76230, Mexico; jm.lopez@cinvestav.mx (J.M.L.-R.); prokhorov@cinvestav.mx (E.P.); kovalenko.yuriy@gmail.com (Y.K.); 3Cinvestav Saltillo, Ramos Arizpe 25900, Mexico; jmendeznonell@cinvestav.mx; 4Department of Sciences & Engineering, Universidad Iberoamericana, San Andrés Cholula 72820, Mexico; mariajose.rivas@iberopuebla.mx; 5Department of Postgraduates Studies and Investigation, Tecnologico Nacional de Mexico, Instituto Tecnológico de Pachuca, Pachuca 42080, Mexico; marisa.mr@pachuca.tecnm.mx (M.M.-R.); erika.ad@pachuca.tecnm.mx (E.O.Á.-D.); 6Immunology and virology Laboratory, Universidad Autónoma de Nuevo León, Monterrey 64450, Mexico; 7Department of Engineering & Technology, Division of Chemical Sciences, FES-Cuautitlan, Universidad Nacional Autónoma de Mexico, Cuatitlan Izcalli 54740, Mexico; 8Department of Molecular Biomedicine, Cinvestav Zacatenco, Ciudad de Mexico 07360, Mexico

**Keywords:** chitosan-gold nanocomposites, HAuCl_4_/sodium citrate relationship, *α*-relaxation, *σ*-relaxation

## Abstract

Nanocomposite engineering of biosensors, biomaterials, and flexible electronics demand a highly tunable synthesis of precursor materials to achieve enhanced or desired properties. However, this process remains limited due to the need for proper synthesis-property strategies. Herein, we report on the ability to synthesize chitosan-gold nanocomposite thin films (CS/AuNP) with tunable properties by chemically reducing HAuCl_4_ in chitosan solutions and different HAuCl_4_/sodium citrate molar relationships. The structure, electrical, and relaxation properties of nanocomposites have been investigated as a function of HAuCl_4_/sodium citrate molar relation. It was shown that gold particle size, conductivity, Vogel temperature (glass transition), and water content strongly depend upon HAuCl_4_/sodium citrate relationships. Two relaxation processes have been observed in nanocomposites; the *α*-relaxation process, related to a glass transition in wet CS/AuNP films, and the *σ*-relaxation related to the local diffusion process of ions in a disordered system. The ability to fine-tune both *α*- and *σ*-relaxations may be exploited in the proper design of functional materials for biosensors, biomaterials, and flexible electronics applications.

## 1. Introduction

Metal nanoparticles exhibit unique optical, electrical, mechanical, and biomedical properties. In this regard, the number of publications related to polymer-metal nanocomposites has recently increased [1,2,3,4,5,6,7,8,9,10,11,12,13,14,15]; various studies report on the use of such nanocomposites in biomedicine, biosensors for protein recognition, and flexible electronics. Most of these reports rely on the proper chemical synthesis of Au nanoparticles; however, the challenge to fine-tune the structure, size, and functionality is still at large. In this regard, the synthesis of gold nanoparticles has been studied for more than 20 years to obtain homogenous sizes and shapes. The preferred method of AuNP synthesis is the well-established Turkevich method, and it is based upon the chemical reduction of HAuCl_4_ in a variety of media. AuNPs are then formulated with a variety of polymer matrices used to fabricate nanocomposites for specific applications; the literature reports the best polymer choices for AuNPs are polyvinyl alcohol (PVA), polyvinyl pyrrolidone (PVP), polyethylene glycol (PEO), and the polysaccharide chitosan (CS) [1,2,3,4,5,6,7,8,9,10,11,12,13,14,15,16]. Moreover, interesting chemical reduction methods to produce functional metal nanoparticles have been discussed in the literature; in this regard, Ag [17], Au/Ag [18], Pt/Pd [19], and biofunctionalized Au nanoparticles [20]. These studies reveal the importance of proper synthesis methods for biosensor applications.

Based upon the above polymer matrix choices for nanocomposite fabrication, chitosan (CS) has emerged as one the preferred ones such that it has been extensively studied [21,22]. In this regard, CS is a natural polymer derivative of chitin which is composed of glucosamine and N-acetylglucosamine unit residues; this polysaccharide displays properties such as biocompatibility, biodegradability, and low toxicity [23,24,25,26]. The cationic nature of chitosan is primarily responsible for electrostatic interactions with metallic nanoparticles such that it can be used as a stabilizer and reducer of gold forming zero-valent nanoparticles [27,28,29,30,31]. It is noteworthy that the chemical reduction of HAuCl_4_ to synthesize AuNPs of specific sizes depends upon the redox agent sodium citrate concentration [29].

Our research group has extensively reported the synthesis of Au NPs by using a modified Turkevich method for a variety of applications [28,30,31,32,33]. However, in the literature, there are no reports about the effect of the HAuCl_4_/sodium citrate concentration relationship on the synthesis and final properties of gold nanoparticles using the Turkevich method in chitosan solutions. Chitosan/gold nanoparticle (CS/AuNP) composites find wide application in biomedicine because gold nanoparticles help promote the protein expression in the keratinocytes process regeneration and glioma cells; it has also been shown to be a tool in the detection and treatment of cells cancer [27,29,34]. Moreover, the development of biosensors requires proper optical, piezoelectric, or electrical responses. Electrochemical biosensors, which convert biological binding into useful electrical signals, have received considerable attention in the past years [5,6]. In most cases, AuNPs dispersed in 3D chitosan matrices served as current conductors [5,6,35]. However, the conductivity mechanism of CS/AuNP nanocomposites has not been properly addressed.

This work aims to provide a robust yet simple chemical synthesis route to fabricate CS/AuNPs nanocomposites; these nanocomposites may find potential applications in biosensors, biomaterials, and flexible electronics. In this regard, in situ AuNPs are synthesized in chitosan solutions with different HAuCl_4_/sodium citrate relationships. We aim to vary this HAuCl_4_/sodium citrate relationship to tune the structural, electrical, and thermal properties using dielectric spectroscopy, thermogravimetric analysis (TGA), and Fourier transformed infrared spectroscopy (FTIR) measurements.

## 2. Materials and Methods

### 2.1. Synthesis of Nanocomposite

Chitosan with medium molecular weight (300,000 g/mol) and 85% of the degree of deacetylation (cat. num. 448877), hydro chloroauric acid (cat. Num. 50790), and sodium citrate (cat. num. S1804) were obtained from Sigma Aldrich^®^ (Lerma, MEX, Mexico) and were used without additional purification.

Seven materials were synthesized with the same molar concentration of HAuCl_4_ and different amounts of sodium citrate (SC). The nanocomposites were synthesized by dissolving 2% of chitosan in a 1% acetic acid solution. After that, a 0.3 mM of HAuCl_4_ solution was added to different amounts of SC to vary millimolar relationships between HAuCl_4_ and SC in the range 0.1 and 5. Then the mixture was heated to 75 °C under magnetic stirring until the solution changed its color to red.

A modified Turkevich chemical reduction method was followed according to a recent study from our group [34]. In this regard, chitosan is a polysaccharide soluble in aqueous acid media below pH 6. At low pH, the amino groups are protonated, allowing the formation of a water-soluble polyelectrolyte. The amount of citrate added to the reaction should not increase the pH of the solution above 6 to overcome unwanted CS precipitation. Solutions were heated to 75 °C under magnetic stirring until they turned red. Films with a thickness of ca. 20 μm were prepared by the solvent-cast method by pouring the final solution into a plastic Petri dish and allowing the solvent to evaporate for 24 h at 60 °C.

20 μm films were prepared to perform impedance spectroscopy measurements; for FTIR in the transmission mode measurements, ca. 10 μm films were prepared.

### 2.2. Characterization Studies

#### 2.2.1. Infrared Measurements and Morphology Analysis

The materials were characterized by infrared spectroscopy. FTIR spectra were obtained between 4000 and 400 cm^−1^ (Perkin Elmer Spectrum1 model, PerkinElmer, Inc. Waltham, MA, USA). All spectra were recorded at 4 cm^−1^ intervals and 16 cm^−1^ times scanning using the transmission technique. CS/AuNPs films morphology was imaged by JEOM JSM-7401F field emission scanning electron microscope (JEOL Inc., Peabody, MA, USA). UV-Vis (UV-Vis spectrometer Agilent 8453, Agilent Technologies, Santa Clara, CA, USA) was used to determine the sizes of the gold nanoparticles by the detection of the maximum absorption band in the visible region.

#### 2.2.2. Thermal Measurements

The amount of free water was determined by Thermogravimetric analysis (TGA) (Mettler Toledo 851e model, Mettler Toledo, Columbus, OH, USA). The measurements were performed with a dry airflow from 25 to 300 °C with a rate of 10 °C/min.

#### 2.2.3. Dielectric Measurements

Dielectric measurements in the frequency range from 40 Hz to 110 MHz were carried out with Agilent Precision Impedance Analyzer 4249A (Santa Clara, CA, USA). The amplitude of the measuring signal was 100 mV. Temperature measurements were performed in the cell in the temperature range from 20 °C to 200 °C using a temperature controller programmed to produce a constant heating rate of 3 °C/min between certain measurements temperature. To remove moisture content in the films, additional measurements were carried out in an in-house vacuum cell [36].

## 3. Results and Discussion

### 3.1. Infrared Spectroscopy

Figure 1 shows the IR spectra of chitosan-gold nanocomposite with a constant concentration of HAuCl_4_ and different amounts of sodium citrate; neat CS, and molar ratios of HAuCl_4_/sodium citrate 0.1, 0.21, 1, 2, and 5. The band at 3300 cm^−1^ is produced by a symmetric stretch of -OH groups. The band at 2880 cm^−1^ is associated with a symmetric stretching of the methyl group, the band present at 1640 cm^−1^ belongs at C=O antisymmetric, from citrate moiety, additionally, the band at 1550 cm^−1^ is assigned to the antisymmetric deformation of NH_3_^+^. The band at 1410 cm^−1^ is produced by the C-N stretch. The stretching band at 1025 cm^−1^ is related to C-O [1,2,3,36].

The salient features of the FTIR spectra are in the vicinity of the 1640 cm^−1^ band, where intensity in the spectrum decreases with decreasing SC concentration. This effect could be attributed to the low efficacy of sodium citrate to reduce gold.

### 3.2. Morphology Analysis

Figure 2a,b show SEM micrographs where a homogeneous distribution of Au nanoparticles is embedded in the chitosan matrix. Using AutoCAD 2007 software, the dimension of nanoparticles can be assessed. Figure 2c shows particle distribution histogram for HAuCl_4_/SC relation equal to 1; this histogram has been obtained from 3 micrographs. Most nanoparticles are 8 to 11 nm (76%). By increasing the HAuCl_4_/SC from 0 to 1 particle size decreases; upon further increase, particle size slightly increases. Similarly, the maximum absorption UV-Vis in the ultraviolet-visible spectrum decreases, then it almost remains constant (Figure 2d). The shift of the maximum absorption band to lower wavelength confirms the decrease in AuNP size. In the conventional Turkevich synthesis of AuNPs where sodium citrate is used as a reducing agent (without chitosan), the increasing SC concentration decreases particle size [37]. It is noteworthy that our method, which uses chitosan as a co-reducing agent, helps reduce the amount of sodium citrate, nanoparticle size decreases. This trend can be explained by the reducing capacity of CS and the presence of free amino and hydroxyl groups in chitosan and its polycationic and chelating properties. The special feature of this polysaccharide enables its use as a stabilizer and reducing agent in the synthesis of gold nanoparticles [36]. Additionally, an excess volume of HAuCl_4_ promotes the nucleation of smaller particle sizes and polydispersity [36].

### 3.3. Thermal Measurements

Thermogravimetric analysis (TGA) was performed to determine free water content. Free water content may be evaluated by the decrease in sample weight during the heating scan. The weight loss at 120 °C is the result of water evaporation. Figure 3 shows the dependence of moisture content as a function of the HAuCl_4_/SC molar relationship. Here, an increase in the HAuCl_4_/SC molar relationship to 1 and a decrease in moisture content are observed. Upon further increasing this HAuCl_4_/SC relationship, free water content slightly decreases, as observed in Figure 3. In this regard, the major change of free water content occurs for HAuCl4/SC molar relationships from 0 to ca. 1 (ca. 17%), while a minimal change of ca. 3% is observed for up to HAuCl4/SC molar relationship of 5.

Additional TGA measurements were carried out on films that were annealed for 30 min at 120 °C with subsequent cooling to room temperature; a second scan was then performed. In such annealing films, the water content was ca. 0.2 wt.%.

### 3.4. Conductivity Measurements

Complex impedance spectra (Z″ versus Z′) for all films exhibit characteristic semicircles at high frequencies and a quasi-linear response at low frequencies (insert in Figure 4). The linear response at low frequencies can be associated with interfacial polarization or metal contact effects [27]. The values of DC resistance R_dc_ have been obtained by fitting the high-frequency semicircle of the impedance spectra before interception with real parts of impedance as depicted in the Figure 4 insert. The corresponding DC conductivity (*σ*_dc_) has been obtained from the equation *σ*_dc_ = d/(R_dc_ × S), where d is the film thickness and S is the area of film.

From Figure 4, one can observe that the nanocomposite’s conductivity increases from 0 up to HAuCl_4_/SC molar relationship of ca. 1.0; for higher HAuCl_4_/SC molar relationship, the conductivity remains almost constant (saturation). This observation qualitatively agrees with the results shown in Figure 2; there is a saturation of both nanoparticle size and conductivity.

### 3.5. Dielectric Measurements

Dielectric measurements were performed using the Agilent Precision Impedance Analyzer 4249 A in the frequency range from 40 Hz to 110 MHz. The amplitude of the measured signal was 100 mV. Temperatures were measured in the cell from 20 °C to 200 °C using a temperature controller programmed to produce a constant heating rate of 3 °C/min between certain measured temperatures. Each sample was kept for 3 min at each temperature to ensure thermal equilibrium. Additional measurements were conducted in a vacuum cell to remove moisture from the films. As-prepared samples were annealed into the vacuum cell before measurements at 120 °C for 1 h, followed by cooling at room temperature in the vacuum. Additionally, a Peltier heating element was used to perform measurements from 0 °C to 100 °C.

Dielectric measurements can provide information about the temperature relaxation processes in the nanocomposites. In polymer-metal nanoparticle composites, both ionic current and interfacial polarization could often mask the real dielectric relaxation processes in the low-frequency range. Therefore, to analyze the dielectric process, the complex permittivity *ε^*^* has been converted to the complex electric modulus *M^*^* by the following equation:(1)$$M^{*}=\frac{1}{\varepsilon^{*}}=M^{\prime }+iM^{\prime \prime }=\frac{\varepsilon^{\prime }}{{\varepsilon^{\prime }}^{2}+{\varepsilon^{\prime \prime }}^{2}}+i\frac{\varepsilon^{\prime \prime }}{{\varepsilon^{\prime }}^{2}+{\varepsilon^{\prime \prime }}^{2}}$$
where $M^{\prime }$ is the real and $M^{\prime \prime }$ the imaginary part of electric modulus, $\varepsilon^{\prime }$ is the real and $\varepsilon^{\prime \prime }$ the imaginary part of permittivity. In this representation, interfacial polarization and electrode contributions are essentially suppressed [27]. The corresponding relaxation time can be calculated by the next relation: *τ* = 1/(2π*f_p_*), where *f_p_* is the peak frequency in the dependence of $M^{\prime \prime }$ on frequency [38].

It is noteworthy that a recent study of our group reported three molecular relaxations in CS-Au nanocomposites based upon dielectric measurements [38]:(1)in the temperature range of 25–70 °C, a nonlinear behavior is revealed,(2)a linear behavior in the temperature range of 70–150 °C, and(3)at temperatures above 160 °C, polymer degradation is triggered.(4)Based upon this analysis, Figure 5 shows the dependencies of the relaxation time on reciprocal temperature based upon the methodology described in [27] for a HAuCl_4_/SC molar relationship of 0.21.

In the temperature range of 25–70 °C the nonlinear relaxation process can be fitted by the Vogel–Fulcher–Tammann (VFT) relationship $\tau =\tau_{0}\exp(\frac{DT_{0}}{T-T_{0}})$, where *T*_0_ is Vogel temperature, *τ*_0_ and *D* are empirical material-dependent parameters.

In the linear relaxation process (80–150 °C), an Arrhenius-type linear dependency was observed ($\tau =\tau_{0}\exp(\frac{E_{a\tau }}{RT})$) for as-prepared films and in the temperature range 22–150 °C for dry films (annealed in vacuum to the temperature of 120 °C).

The *σ*-relaxation appears when there is dominance of ionic conductivity in CS-Au nanocomposites and in neat chitosan and most polysaccharides [36]. This relaxation appears due to ion migration which is responsible for additional dielectric polarization in amorphous and electrically inhomogeneous systems.

The nonlinear *α*-relaxation, which is related to a glass transition, appears in wet CS-Au nanocomposites. Figure 6 shows the Vogel temperature *T*_0_ as a function of the HAuCl_4_/SC molar relationship; an analogous behavior for the glass transition temperature of such nanocomposites is expected at 50–70 K higher than *T*_0_ [38].

Figure 6 shows the existence of excess HAuCl_4_ is responsible for the “anomaly” dependence of Vogel temperature. The Vogel temperature *T*_0_ is the apparent activation temperature of the *α*-relaxation in many polymers; *T*_0_ is usually 50–70 K lower than glass transition temperature [34].

## 4. Discussion

In this work, CS/Au nanocomposites have been prepared with different HAuCl_4_/SC molar relationships; this means that some HAuCl_4_ will be complexed in the films and can participate in the observed ionic DC conductivity. By decreasing the concentration of SC (an increase in HAuCl_4_), there is an increase in the film’s conductivity. When the HAuCl_4_/SC molar relationship is higher than 1.0, a competitive reducing capacity of chitosan helps compensate for the reduced amount of SC. The ability of chitosan to help reduce HAuCl_4_ is due to the presence of free amino and hydroxyl groups. This effect is responsible for saturation in the particle size (Figure 2c), conductivity (Figure 4), water absorption (Figure 3), and Vogel temperature (Figure 6).

The presence of water absorption in neat chitosan films promotes a plasticizing effect; for this reason, in the samples with lower moisture content, the Vogel temperature is lowered [38]. Nevertheless, in the case of CS/AuNP nanocomposites, the Vogel temperature decreases with increasing HAuCl_4_/SC molar relationship even though moisture content decreases (see Figure 3 and Figure 6). According to our previous work [36], gold nanoparticles at low HAuCl_4_/SC molar relationship attach to chitosan through hydrogen bonds between complexed sodium and the amino groups of chitosan. Upon increasing the HAuCl_4_/SC molar relationship, complexed sodium decreases, and the interaction between AuNPs and chitosan could be due to electrostatic forces (due to the polarization of nanoparticle surface. This effect is responsible for the decreasing of Vogel and glass transition temperatures. A similar conclusion can be drawn from the dependencies of particle size (Figure 2c) and the conductivity at HAuCl_4_/SC molar relationships higher than 1.0.

It is noteworthy that by varying the HAuCl_4_/SC molar relationship, the nanocomposite’s water absorption, conductivity, and the glass transition temperature can be fine-tuned. Samples prepared with low HAuCl_4_/SC molar relationships have higher complexed sodium molecules. These molecules form hydrogen bonds with the polymer matrix, and water can attach to the OH group of chitosan. On the other hand, the films with high HAuCl_4_/SC molar relationships have less complex sodium molecules such that the quantity of hydrogen bonds is also lower. Additionally, any excess of HAuCl_4_ can potentially be reduced by the chitosan’s reducing capabilities. Although the reducing process by chitosan is not well understood, it is possible that the OH groups can act as reducing groups in the formation of nanoparticles [39]. Au nanoparticles that are charged superficially can attach to NH_3_^+^ groups by electrostatic forces. The nanocomposite water absorption is lower for materials with higher HAuCl_4_/SC molar relationships due to the decreasing hydrogen bonding capacity.

## 5. Conclusions

CS/AuNP thin films have been synthesized by chemical reduction of HAuCl_4_ in the presence of sodium citrate (SC) and chitosan solutions. The structure, conductivity, and relaxation properties of CS/AuNPs films have been investigated as a function of the HAuCl_4_/SC molar relationship. Our results show that HAuCl_4_/SC molar relationship affects the AuNP sizes because of the additional chitosan reducing agent capabilities. A descriptive model for understanding the reaction between HAuCl_4_, sodium citrate chitosan, and acetic acid has been proposed. We have shown that an excess of HAuCl_4_ is responsible for dependencies of conductivity, Vogel temperature, and water absorption in CS/AuNPs nanocomposites. The value of HAuCl_4_/SC molar relationship of ca. 1.0 is a molar relationship threshold. At higher HAuCl_4_/SC molar relationships, the reducing capacity of chitosan leads to saturation of free HAuCl_4_ amount and subsequently to saturation in all properties of CS/AuNP nanocomposites.

## Figures and Tables

**Figure 1 polymers-14-00788-f001:**
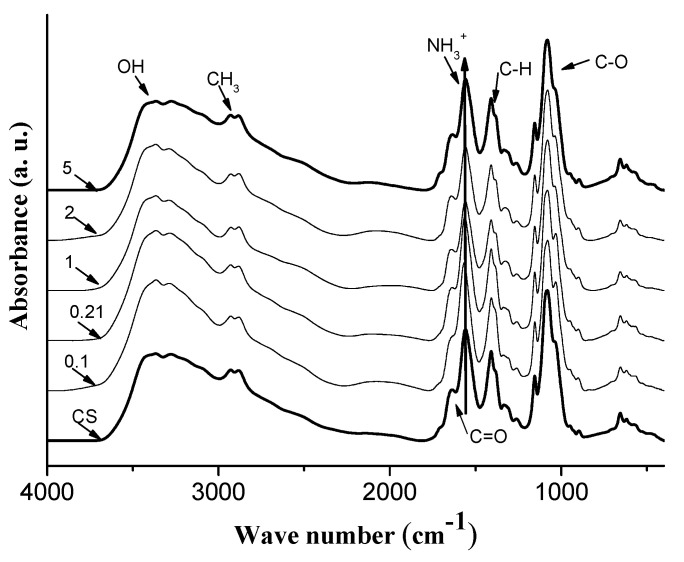
FTIR spectra of neat chitosan (CS) and CS/AuNPs composites with HAuCl_4_/SC (M/M, molar ratio) relationships of 0.1, 0.21, 1, 2, and 5. Note the spectral region of ca. 3200 to 3500 cm^−1^ reveals a subtle widening of the vibration bands that are traceable to H-bonding.

**Figure 2 polymers-14-00788-f002:**
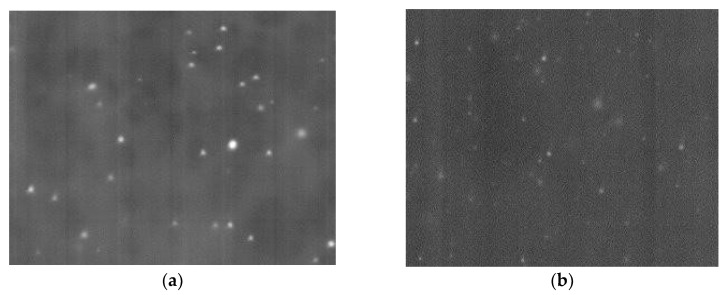
SEM micrographs of CS/AuNPs films with HAuCl_4_/SC molar ratios of (**a**) 0.21 and (**b**) 1.0. Measurement conditions: Secondary electrons (SE), accelerating voltage 5.0 KV, magnification ×100,000, working distance (WD) 8.6 mm; (**c**) Histogram of nanoparticle distribution for HAuCl_4_/SC relation equal 1.0; (**d**) Dependence of maximum absorption in the ultraviolet visible spectrum on HAuCl_4_/SC molar ratio.

**Figure 3 polymers-14-00788-f003:**
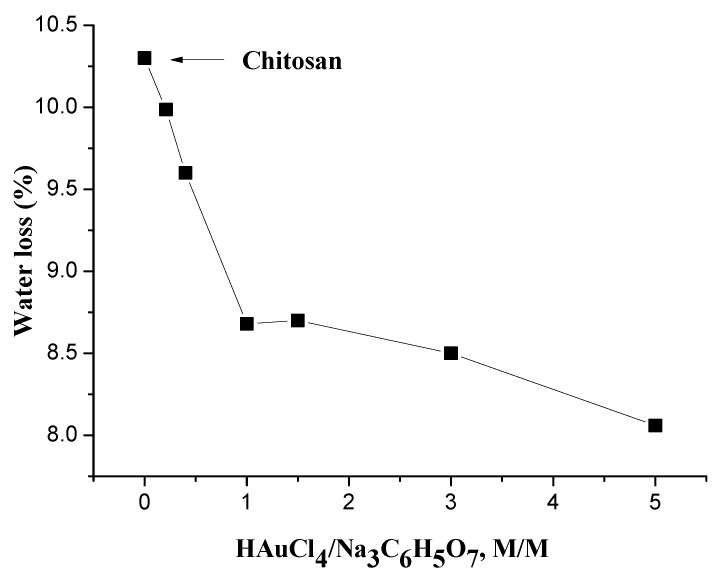
Dependence of water content on HAuCl_4_/SC molar relationship.

**Figure 4 polymers-14-00788-f004:**
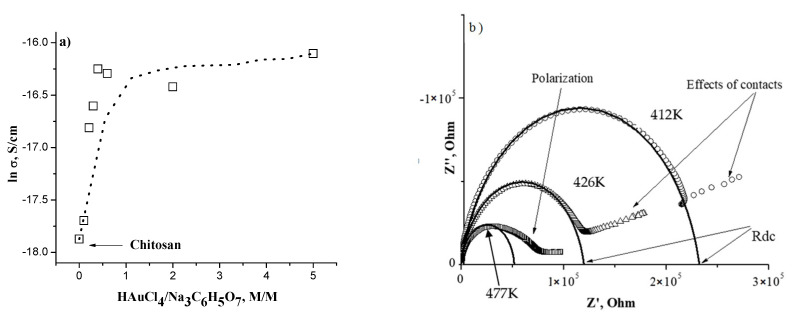
(**a**) Dependence of DC conductivity on HAuCl_4_/SC molar relationship. (**b**) Insert shows impedance spectra obtained at the temperature indicated on the graph.

**Figure 5 polymers-14-00788-f005:**
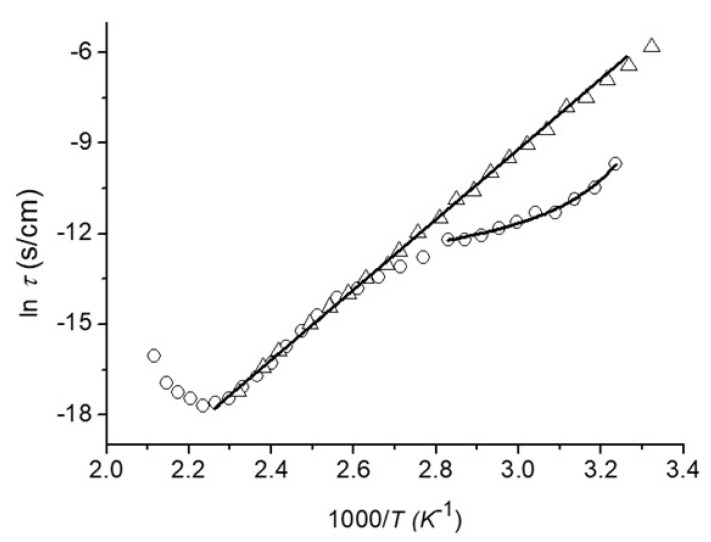
Dependence of the relaxation time on the reciprocal temperature in as-prepared (open circles) and annealed films (open triangles) for HAuCl_4_/SC molar relationship of 0.21. Note an Arrhenius-type linear fit associated with *σ*-relaxation (activation energy ca. 103.2 kJ/mol) and the nonlinear VFT fit associated with an *α*-relaxation (glass transition).

**Figure 6 polymers-14-00788-f006:**
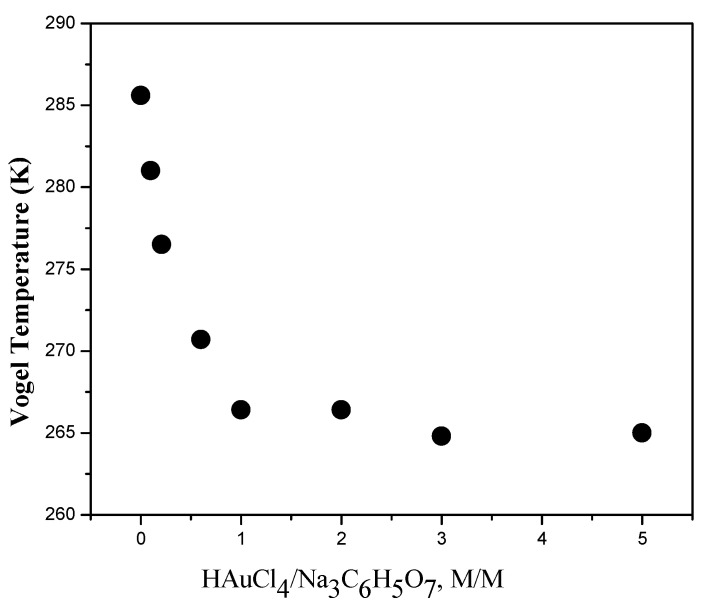
Dependence of Vogel temperature on HAuCl_4_/SC molar relationship. Note that for most amorphous polymers, the glass transition is 50–70 K higher than *T*_0_.

## Data Availability

The data presented in this study are available on request from the corresponding author.

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
