# Peer review of "Tuning HAuCl4/Sodium Citrate Stoichiometry to Fabricate Chitosan-Au Nanocomposites"

_polymers, 2022, doi:10.3390/polym14040788_

Round 1
Reviewer 1 Report
The manuscript titled, “Tuning HAuCl4/Sodium citrate stoichiometry to fabricate Chitosan-Au nanocomposites” by Luis et al did a detailed study on engineering the HAuCl4/Sodium citrate molar relationships. The present manuscript is interesting and author’s presentation and results discussion part is good. However, authors should address the following minor issues in the manuscript before being publication in Polymers.
- In the introduction section, the authors should include some more recent achievements on metal nanoparticles and their applications from the literatures.
- In FE-SEM images, the author should confirm the distribution of nanoparticles and its uniformity throughout the sample. In the image shown, the particles are seemed to be concentrated in a few spots.
- The author should recheck for spelling mistakes and grammatical errors in some part of the manuscript.
- In order to confirm the structural changes, the author should check the cross sectional FE-SEM images.
- In FTIR analysis, all the spectra remain the same even after adding the composite. The authors should explain the inference in detail.
- In the thermal measurements, the authors mentioned that “there was no change in free water content particles upon the addition of the HAuCl4/SC relationship” but the data in Fig.3 shows a decline in the free water content values.
- The authors should explain what the symbol represents in the Vogel-Fulcher-Tammann (VFT) relationship on page 7.
- Several closely related references improve biosensors and biomaterial applications. These works should be introduced to the broader readers in suitable place, providing more information, such as DOI: 10.1039/C4AY01183J; 10.1021/ie3022797; 10.1039/C8NJ02782J ; 10.5012/bkcs.2011.32.12.4171.
Reviewer 2 Report
A method for fine-tune properties of a chitosan-Au nanocomposite is described. I recommend the manuscript for publishing, but I have some questions/comments:
- How was the film thickness estimated and how the FTIR spectra were recorded? It seems that 20 µm thickness can be quite large for a transmission spectrum. Are the spectra shown in Fig.1. normalized? To which band?
- FTIR spectra should give an information about the postulated hydrogen bonding between citrate moieties and chitosan.
- Page 4 “Additionally, an excess volume of HAuCl4 promotes a nucleation of smaller particle size and polydispersity” while in Fig.2d one may notice a small increase of Ag particles for HAuCl4/Na3C6H5O7 >1. It correlates with conductivity (Fig.4a). Please explain.
- Figure 3 – what is the point of this? Please explain more clearly. The molar amount of H2O introduced during the preparation of the nanocomposite varies and it may cause differences during the evaporation of water over 24 hours.
- Page 8 “Samples prepared with low HAuCl4/SC molar relationships have higher complexed sodium molecules. Theses molecules form hydrogen bonds with the polymer matrix and water can attach to OH group of chitosan.” – it is misleading. What sort of hydrogen bonds do you mean exactly?
- Page 9 “Au nanoparticles that are charged superficially can attach to NH3+ groups by electrostatic forces.” – could you comment on the type of charge Au nanoparticles obtained in this system have?
- Page 9 “it is possible that the OH groups can act as reducing group in the formation of nanoparticles.” – please add a literature reference.
- Kelvin degrees and Celsius degrees are used – please unify the system. Moreover zero in superscript (0C) is used instead of °C.
- Page 7 is “nonlinear a-relaxation” – should be “nonlinear α-relaxation”
Author Response
Please see the attachement

Round 2
Reviewer 2 Report
The manucript has been improved and almost could be published in its present for except that The degree C and K problem has not been solved completely. I would suggest using Kelvins since it would match Figs 5 and 6. Please remember to correct the temperature indicators in Fig. 4b.
Author Response
We have corrected units in Fig. 4b from ºC to K.
Sincerely,
Gabriel